🔓 | **Open Peer Review** | Antimicrobial Chemotherapy | Research Article

# The impact of aztreonam–clavulanic acid exposure on gene expression and mutant selection using a multidrug-resistant *E. coli*

Tongtong Lin,[1,2] Jiayuan Zhang,[1] Shuo Diao,[1] Jinke Yan,[1] Kexin Zhang,[1] Jichao Cao,[1] Junyi Huang,[1] Yaohai Wang,[1] Zhihua Lv,[1] Xiaopeng Shen,[3] Sherwin K. B. Sy,[4] Michael Lynch,[5] Hongan Long,[1,2] Mingming Yu[1]

**ABSTRACT**  Multidrug-resistant *Escherichia coli* poses a significant threat to the healthcare system by causing treatment failure in infected patients. The use of a beta-lactam in combination with a beta-lactamase inhibitor has been shown to be an effective strategy to solve this problem. *In vitro* antimicrobial susceptibility experiments have demonstrated the antimicrobial activity of aztreonam and clavulanate. In this investigation, we conducted a transcriptomic analysis to reveal the downstream differential gene expression in *E. coli* ymmD45 (a strain newly isolated and found to carry the New Delhi metallo-β-lactamase gene) following exposure to aztreonam and clavulanate separately, as well as their combination. Differential gene expression, pathway enrichment, and gene network analyses demonstrated the polygenic nature of the response to the combination treatment, which suppressed the expression of pivotal virulence genes, disrupted two-component regulatory systems for bacteria to resist external stress, and interfered with the formation of the cellular membrane. Results from single-step mutant selection combined with deep whole-genome sequencing also revealed the spontaneous origin of the resistance mutations and confirmed action mechanisms during the combination treatment. Our study contributes valuable insights into the impact of antibiotic exposure on gene expression, laying the groundwork for understanding antibiotic resistance development in the treatment of multi-drug resistant infections through *in vitro* studies.

**IMPORTANCE**  Multidrug-resistant *Escherichia coli* is a major challenge in treating infections effectively. Aztreonam and clavulanate combination is promising in combating these resistant bacteria. By investigating the antimicrobial activity of aztreonam and clavulanate using transcriptomic analysis and mutant selection, this research sheds light on the mechanisms underlying antibiotic resistance and the effectiveness of combination therapies. The findings highlight how this particular antibiotic combination suppresses virulence genes, disrupts bacterial regulatory systems, and interferes with cellular functions critical for resistance. Moreover, the study lays the groundwork for understanding antibiotic resistance development in the treatment of multi-drug resistant infections through *in vitro* studies, offering insights that could inform future strategies in clinical settings. Ultimately, our findings could guide the development of better treatment strategies for multidrug-resistant infections, improving patient outcomes and helping to manage antibiotic resistance in healthcare.

**KEYWORDS**    antibiotic combination, resistance mutations, evolution, mutagenesis

Address correspondence to Mingming Yu, yumingming@ouc.edu.cn, or Hongan Long, longhongan@ouc.edu.cn.

Tongtong Lin and Jiayuan Zhang contributed equally to this article. Author order was determined alphabetically.

Author Hongan Long is one of the Editorial Board members, but he was not involved in the journal's review of, or decision related to, this manuscript.

A growing proportion of clinical isolates in humans exhibit resistance to drugs or display multidrug-resistant characteristics, posing a significant challenge to

effective disease treatment (1, 2). Many of these challenges have arisen due to the emergence and dissemination of multidrug-resistant gram-negative bacteria (MDR-GNB), which have developed resistance to all available therapeutic agents, resulting in a paradigm shift in treatment approaches (3, 4). Globally disseminated MDR *Escherichia coli*, one predominant species among MDR-GNBs, poses a significant and pervasive menace to the healthcare infrastructure (5, 6). MDR *E. coli* has the potential to induce conditions, such as bacteremia, sepsis, enteric foodborne illnesses, gastroenteritis, and nosocomial pneumonia, in global human populations (7–10). As a result, effective approaches to controlling bacterial infections are urgently needed. One such approach is the use of antibiotic combinations to combat the MDR phenotype effectively (11, 12).

Numerous factors need to be considered in the selection of antibiotics. Aztreonam, a monobactam, stands out due to its distinct spectrum of activity, which is confined to enteric gram-negative bacilli, coupled with its advantageous toxicity profile (13–15). Metallo-β-lactamase (MBL)-producing gram-negative bacilli exhibit the capacity to hydrolyze all β-lactam compounds, except for aztreonam, although their presence often coincides with other β-lactamases that can diminish aztreonam's effectiveness (16, 17). Clavulanate was the first clinically useful broad-spectrum β-lactamase inhibitor to be described in the literature (18, 19). Functioning as an irreversible 'suicide' inhibitor, clavulanate targets both intracellular and extracellular β-lactamases, exhibiting concentration-dependent and competitive inhibitory properties (20, 21).

Previous studies have demonstrated that the combination of avibactam and aztreonam shows promise against MBL-producing pathogens (22). This combination restores the *in vitro* activity and *in vivo* efficacy of aztreonam by inhibiting the prevalent co-carriage of non-MBL β-lactamases (23, 24). Additionally, the combined utilization of amoxicillin–clavulanate with aztreonam has been explored, but our prior investigations observed a nearly equivalent *in vitro* bactericidal effect between aztreonam–clavulanate and aztreonam–amoxicillin–clavulanate against MDR *E. coli* (25, 26). These findings suggest limited advantages in incorporating amoxicillin alongside aztreonam–clavulanate. Our current study focuses on elucidating the impact on gene expression of the aztreonam–clavulanate combination, which is beneficial to its clinical application.

In this study, we conducted in-depth research using an MDR *E. coli* strain, ymmD45. Through *in vitro* antimicrobial susceptibility experiments, we determined the minimum inhibitory concentration (MIC) values for aztreonam, clavulanate, and the aztreonam–clavulanate combination. To elucidate the impact of the aztreonam–clavulanate combination, we performed RNAseq-based differential gene expression, pathway enrichment, and gene network analyses of *E. coli* ymmD45 following exposure to them. Additionally, we screened resistant strains surviving a high concentration of aztreonam–clavulanate combination exposure, identifying the putative resistance mutations that they carried, with the aim of tracking the origins of resistance mutations and their potential functional mechanisms (Fig. 1).

## MATERIALS AND METHODS

### Clinical isolates, antibiotics and media

*E. coli* ymmD45 was isolated from the sputum of patients with pulmonary infection at the Affiliated Hospital of Qingdao University. Next-generation sequencing and the CARD database were employed to identify the antibiotic resistance genes present in this strain. For susceptibility testing, *E. coli* strains ATCC 25922 and 35218 (obtained from Shanghai Aiyan Biotechnology Co., Ltd., Shanghai, China; cat. nos.: CL7080 and CL1971) were used as quality control strains. Analytical-grade clavulanate and aztreonam (ordered from Shanghai Macklin Biochemical Co., Ltd., Shanghai, China; cat. nos.: P890203 and A801653) solutions were prepared according to Clinical and Laboratory Standards Institute (CLSI) guidelines. Luria–Bertani (LB) agar or broth (Solarbio; cat. nos.: L1015 and L8291, respectively) was used for cell culturing during transferring, freezing, and DNA extraction.

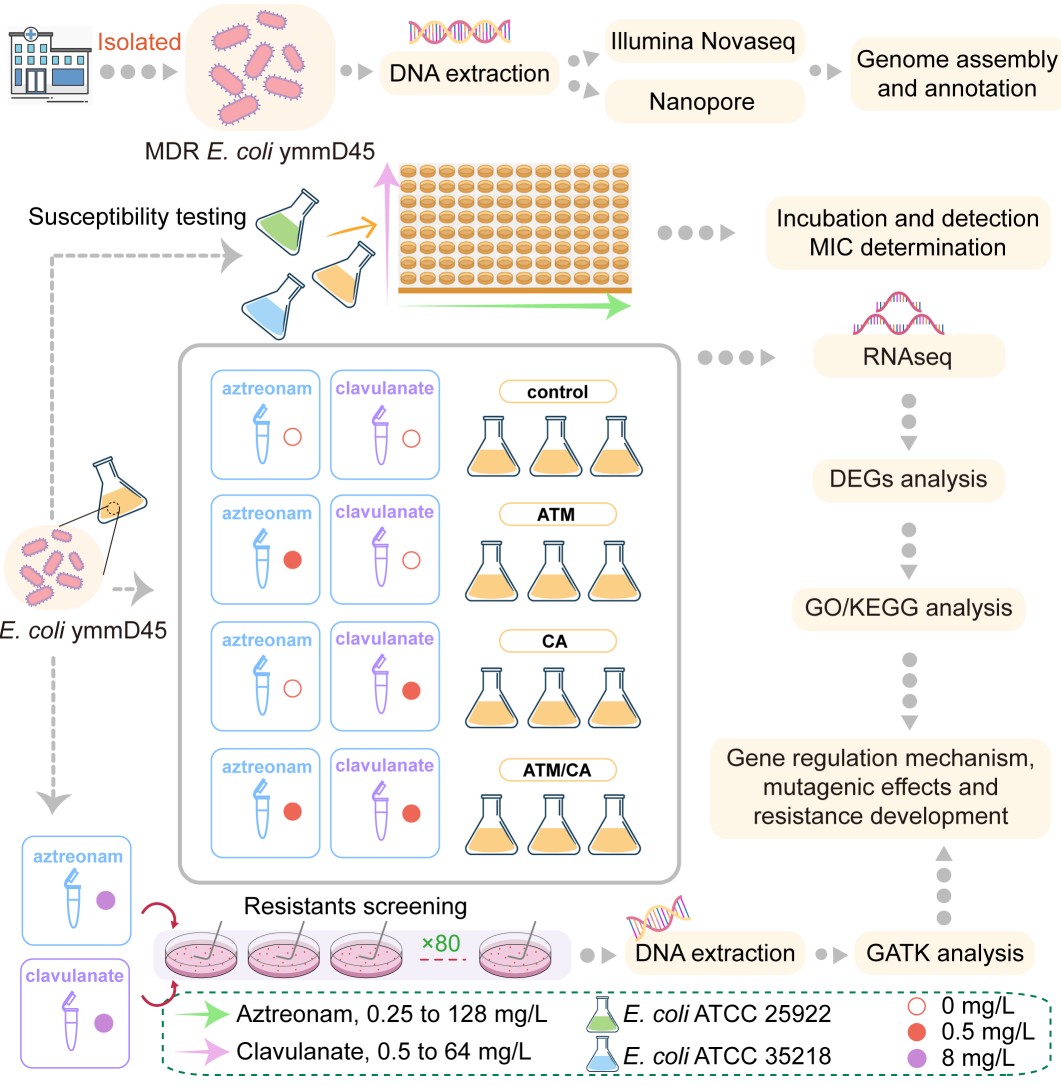

**FIG 1** Experimental design and analytical details of this study.

## Susceptibility testing

According to the CLSI guidelines, a sterile 96-well microdilution plate was used to determine the MIC of aztreonam and clavulanate, either individually or in combination, against the clinical isolate ymmD45. Antibiotic pre-treated plates were prepared, with each well receiving 100 μL of antibiotics at exponentially increasing concentrations. The concentration ranges for aztreonam and clavulanate were 0.25 to 128 and 0.5 to 64 mg/L, respectively. *Escherichia coli* strains were prepared at 0.5 McFarland density and then diluted in each well of the 96-well plate to a final density of $5 \times 10^5$ CFU/mL. The plates were incubated at 37°C for 24 h, and bacterial growth was observed in each well. The MIC was defined as the lowest drug concentration at which bacterial growth was completely inhibited. The growth curves of ymmD45 exposed to 1× MIC were also obtained. The analysis was performed in triplicate.

## Single-step mutant screening combined with deep whole-genome sequencing

From a single ancestral colony of *E. coli* ymmD45, we first cultured it for ~16 h in LB broth to an optical density at 600 nm (OD600) of ~1.5 at 37°C. We then did serial dilution and incubated ~1 × 10⁵ ancestral *E. coli* ymmD45 cells on ~50 LB agar plates with

8 mg/L of aztreonam/clavulanate (16× MIC). Plates were cultured for 24 h at 37°C, and 80 resistant colonies were screened finally for DNA extraction and Illumina sequencing. To evaluate the stability of the mutants, we recovered all 80 resistant colonies from the frozen samples, streaked them three times onto blank LB agar, and then streaked onto LB agar plates with 8 mg/L of aztreonam/clavulanate (16× MIC). Upon incubation, we observed that only 15 strains, exactly the ones with BPSs or indel mutations detected after deep whole-genome sequencing, were capable of growth under these conditions.

## Genomic DNA extraction, library construction, and genome sequencing

For *de novo* genome assembly, the *E. coli* ymmD45 genomic DNA was sequenced using a combination of Nanopore PromethION and Illumina NovaSeq 6000 platforms. For long-read sequencing, the genomic DNA of ancestral *E. coli* ymmD45 was extracted using optimized SDS extraction method. The DNA was checked for purity on a NanoDrop (Thermo Scientific, USA), and DNA concentrations were measured using a Qubit 3.0 fluorometer (Life Technologies, Carlsbad, USA). DNA libraries were constructed using an SQK-LSK110 Ligation Kit with the standard protocol. The purified library was loaded onto a R9.4.1 Flow Cell and sequenced using a PromethION sequencer (Oxford Nanopore Technologies, Oxford, UK) with 48 h runs at Wuhan Benagen Technology Company Limited (Wuhan, China).

Genomic DNA of 80 *E. coli* ymmD45 aztreonam–clavulanate-resistant colonies and ancestral *E. coli* ymmD45 was prepared with the MasterPure Complete DNA and RNA Purification Kit (cat. no. MC85200) based on provided directions, after which the concentration and purity of the resultant DNA were assessed using the Qubit fluorometer (Life Technologies, Carlsbad, CA, USA) with 1× dsDNA HS Assay Kit (cat.: 12642ES76, Wuhan Yeasen Biotechnology, Co., Ltd.) and a Nano-300 spectrophotometer (Allsheng, Hangzhou, China). We constructed DNA libraries by following Li et al. (27), with an insert size around 300 bp, and then paired-end sequenced on an Illumina Novaseq 6000 platform provided by Beijing Novogene Genomics Technology Co., Ltd. (Beijing, China).

## RNAseq of *E. coli* ymmD45

From a single ancestral colony of *E. coli* ymmD45, we first cultured it for ~4 h in 100 mL LB broth to an OD600 of ~0.5 at 37°C. The liquid culture was then split to establish four groups (control, ATM–aztreonam only, CA-clavulanate only, and ATM/CA-clavulanate and aztreonam combination), with each group containing three replicates, 5 mL for each replicate. The three replicates in the group CA were with 0.5 mg/L clavulanate; 0.5 mg/L aztreonam for group ATM; those in the group ATM/CA were with a clavulanate and aztreonam concentration of 0.5 mg/L; and the control group without any antibiotics added (the MIC of ATM/CA combination against *E. coli* ymmD45 is 0.5 mg/L). Here, we used 0.5 mg/L as the aztreonam- or clavulanate-alone concentration in RNAseq, as a control to compare with the combination of 0.5 mg/L ATM and 0.5 mg/L CA. This was done to identify differential gene expressions associated with the enhanced effects of the combination. Cells in 12 tubes were cultured for ~1.5 h to OD600 ~1.0 at 37°C. We then transferred 1.8 mL of each sample to 2.0 mL Eppendorf tubes. Total RNA was then extracted using the MasterPure Complete DNA and RNA Purification Kit (cat. no. MC85200). The concentration and purity of the RNA were measured with a Qubit 3.0 fluorometer (Life Technologies, Carlsbad, CA, USA) and a micro-volume spectrophotometer instrument (Nano-300), respectively. Sequencing libraries were generated using NEBNext Ultra RNA Library Prep Kit for Illumina (NEB, USA, Catalog #: E7530L) following the manufacturer's recommendations and indexes were added for multiplexing. Library quality was assessed on the Agilent 5400 system (Agilent, USA) and quantified by qPCR (1.5 nM). The qualified libraries were pooled and sequenced on an Illumina Novaseq 6000 platform provided by Beijing Novogene Genomics Technology Co., Ltd. (Beijing, China).

## Genome *de novo* assembly and annotation of *E. coli* ymmD45

Nanopore sequencing yielded 508,964 raw reads, ~3.41 Gbp with the longest read of 137,042 bases for *E. coli* ymmD45. Raw nanopore data were filtered by NanoFilt (28), and the reads with quality score below Q10 and sequence length less than 1000 bp were filtered out. The Illumina clean reads were obtained by removing reads containing adaptors, reads containing poly–N, and low-quality reads from ~2.68 Gbp raw reads using fastp (v-0.20.0) (29). After filtering, 2.79 Gbp of Nanopore reads along with 2.37 Gbp clean Illumina short paired-end reads was retained for assembling. The *de novo* assembling of *E. coli* ymmD45 was done with Unicycler (v0.5.0) (30) using both Nanopore long reads and Illumina short reads. Antibiotic resistance genes (ARGs) were annotated by alignment against the CARD using RGI (v5.1.1) (31). Genome assembly was assessed by QUAST (v–5.2.0) (32) and BUSCO (v–5.4.3) (33). The genome annotation was done with the Prokaryotic Genome Annotation Pipeline at the National Center for Biotechnology Information (NCBI) (34). CRISPR sequences were detected using CRISPRFinder (35). PHASTEST was used for searching prophages in the genomes (36, 37). Visualization was based on the TBtools-II bioinformatics platform (38).

## Mutation analyses and statistics

For paired-end raw reads of 80 *E. coli* ymmD45-resistant colonies, we used Trimmomatic–0.38 (39) to trim off adaptors and filter out low-quality data, then mapped the clean reads to the *de novo* assembled genome using BWA (v–0.7.17) MEM (JASMQD000000000, NCBI) (40). Nine samples were removed because of <20× depth of sequencing coverage or mapping rate < 70% to the reference genome. Duplicate reads were removed using Picard tools (v-2.17.2). Base pair substitutions (BPSs) and indels were discovered using standard hard filtering parameters described by the HaplotypeCaller module of GATK (v-4.1.2) (41, 42). Because there might be hotspot genomic sites leading to bacterial resistance, we allowed shared mutations between different lines. Mutation curation was done using the Integrative Genomics Viewer, IGV (v-2.8.12) (43).

## Differential gene expression and pathway enrichment analyses

For RNAseq data of *E. coli* ymmD45, clean reads were obtained by removing reads containing adaptors, reads containing poly–N, and low-quality reads from raw data using fastp (v-0.20.0) (29). Indexing of the reference genome was done using Hisat2 (v-2.1.0), and then paired-end clean reads were mapped to the *de novo* assembled genome JASMQD000000000 (NCBI GenBank) (44). Next, sam files were sorted and converted to bam files by SAMtools (v-1.9) (45). Then, we used Stringtie (v–1.3.7) (46) to assemble the transcriptome and predict new genes, and differential gene expression (DGE) analysis of the two conditions was performed using the DESeq2 R package (1.16.1) (47). The construction of the Gene Ontology (GO) term database and GO enrichment analysis were performed using clusterProfiler (v-4.0) (48). Gene enrichment analyses and functional annotation were achieved by DAVID (v2023q1) (49). The online STRING database (version 12.0) was used for generation of protein interaction (50). All statistics and illustrations were done with R packages ggplot2, ggpubr, dplyr, forcats, gridExtra, viridis, and MASS in R (v-4.0. 2) (51).

## RESULTS

### *De novo* assembly of *E. coli* ymmD45 based on long-read sequencing

Reliable genome and transcriptome analyses necessitate the availability of a high-quality reference genome. In the case of *E. coli* ymmD45, sourced from a hospital setting, the absence of a high-quality reference genome prompted us to initiate the construction of a *de novo* assembly for this particular strain (Fig. 1). The genome analysis of *E. coli* ymmD45 revealed the presence of nine distinct contigs, housing a total of 5,887 genes (Fig. 2; Table 1). There are 85 ARGs annotated in the genome of *E. coli* ymmD45 (Table S1), and five β-lactamase genes were found (Table 2) (22). The genome size is 5,975,948 bp,

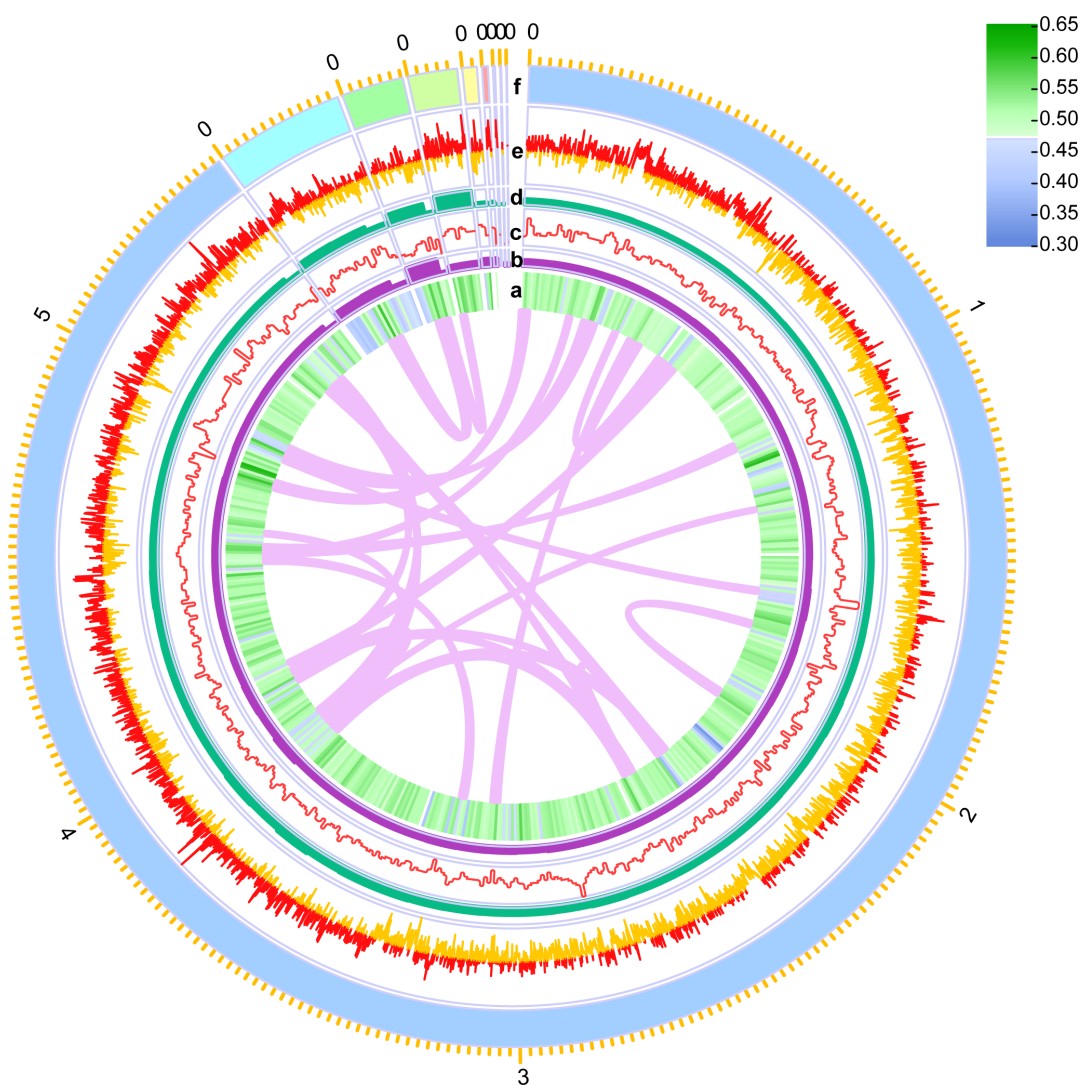

**FIG 2** Summary of the genomic structure of *E. coli* ymmD45 from the inside outward: genomic GC ratio (a), Illumina sequencing depth (b), gene density (c), nanopore sequencing depth (d), genomic GC bias (e), and the nine contigs of the genome (f, numbers are in Mbp). Innermost pink lines show gene synteny within the genome.

exhibiting a GC content of 50.44% (Fig. 2; Table 1) with a BUSCO score of 100. Additionally, we unveiled nine intact prophage sequences spanning 26.6 to 52.4 Kbp in length (Fig. S1; Table S2), alongside the detection of 10 CRISPR sequences comprising two complete and eight incomplete elements (Table S3). The GC content of 10 CRISPR regions are higher than that of the whole genome.

### *In vitro* antimicrobial susceptibility

We first assessed the antibacterial efficacy of the aztreonam–clavulanate combination through susceptibility testing. The MICs for control strains *E. coli* ATCC 25922 and ATCC 35218 were both found to be below 0.5 mg/L for aztreonam (Fig. 1 and 3A; Table 2). These tests met the CLSI standards for accuracy, reliability, and reproducibility in laboratory testing. Notably, *E. coli* ymmD45 is host to an array of resistance genes, including $bla_{NDM-13}$, $bla_{TEM-1B}$, $bla_{TEM-141}$, $bla_{CTX-M-55}$, and $bla_{OXA-1}$, rendering it resistant to aztreonam or clavulanate in isolation (MIC = 64 or 32). However, the synergistic effects observed upon the combined administration of clavulanate with aztreonam are of considerable significance (Fig. 3A; Fig. S2; Table 2). This combinatory

**TABLE 1** Summary of the *de novo* assembly of *Escherichia coli* ymmD45, with *N* as the number of gaps in the assembly.

| Genomic features | E. coli ymmD45 |
| --- | --- |
| Contigs | 9 |
| Largest contig | 5,449,184 |
| Total length | 5,975,948 |
| GC (%) | 50.44 |
| N50 | 5,449,184 |
| N per 100 kb | 0 |
| BUSCO score | 100 |
| Predicted genes (total) | 5,887 |
| CDSs (total) | 5,770 |
| Coding genes | 5,536 |
| Genes (RNA) | 117 |
| rRNAs (5S) | 8 |
| rRNAs (16S) | 7 |
| rRNAs (23S) | 7 |
| Number of tRNAs | 87 |
| ncRNAs | 8 |
| Pseudogenes | 234 |
| CRISPR arrays | 2 |

approach also results in a reduction of the MIC of aztreonam below its clinical breakpoint.

## *E. coli* ymmD45 could effectively regulate intracellular activity when exposed to aztreonam or clavulanate alone

*E. coli* ymmD45 exhibits a multidrug-resistant profile characterized by the presence of $bla_{NDM-13}$ and various other determinants that confer resistance to aztreonam. In our *in vitro* antimicrobial susceptibility assay, *E. coli* ymmD45 displayed resistance to aztreonam in isolation, with MICs exceeding 8 mg/L. To systematically elucidate the impact of aztreonam and clavulanate exposure on the gene expression profile of *E. coli* ymmD45, we conducted RNAseq on control samples and those exposed with aztreonam or clavulanate (Fig. 1). Sample-level quality control was assessed through principal component analysis (PCA) and heatmaps (Fig. 3B and C; Table S4). Subsequently, we performed DGE analyses, employing GO and KEGG pathway enrichment analyses to discern alterations in gene expression patterns. Notably, under aztreonam (0.5 mg/L) or clavulanate exposure (0.5 mg/L), we identified 734 and 1,333 up-regulated (*p*-adjust ≤ 0.05, $\log_2$FoldChange > 1) differentially expressed genes (DEGs), alongside 171 and 414 downregulated (*p*-adjust ≤ 0.05, $\log_2$FoldChange < −1) DEGs when compared with the control (Fig. 3D and E; Table S5 and S6).

In accordance with the outcomes of GO and KEGG analyses, aztreonam exposure resulted in a substantial upregulation of genes associated with cellular membrane formation, encompassing cell membrane, plasma membrane (comprising 399 up-regulated DEGs). Concurrently, this exposure induced an elevation in various cellular processes, including overall biochemical metabolism, particularly phosphorylation and galactose metabolism. It also heightened activities related to flagellar assembly, flagellum-dependent cell motility, transmembrane transporter functions, and bacterial chemotaxis pathways (Fig. S3A; Table S7). Conversely, exposure to aztreonam exhibited inhibitory effects on cytoplasmic activity, ribosome assembly, RNA binding (including tRNA and rRNA), and cationic antimicrobial peptide (CAMP) resistance in *E. coli* ymmD45 (Fig. S3B; Table S7).

In the case of clavulanate exposure, a similar pattern emerged with a noteworthy upregulation of genes associated with cellular and plasma membranes. This exposure

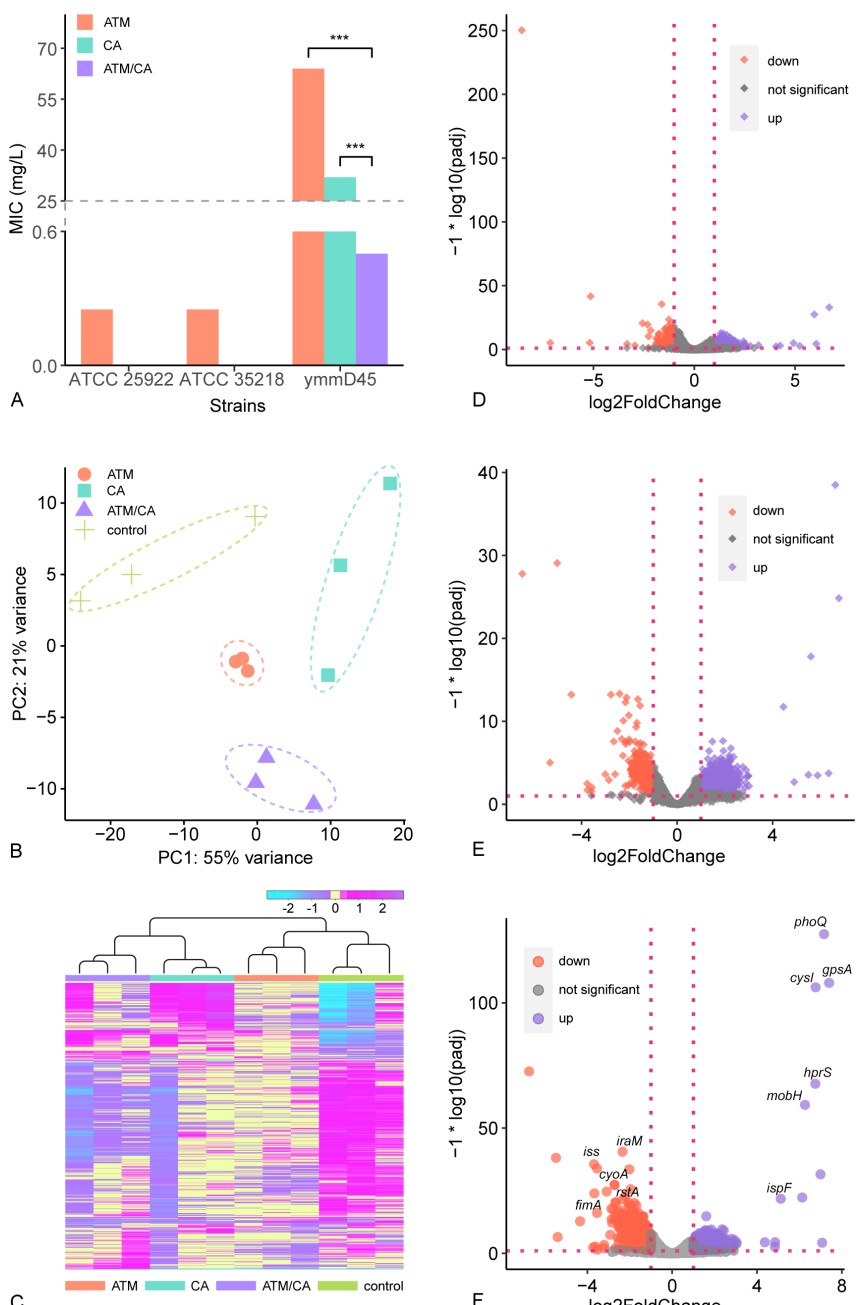

**FIG 3** Resistance and gene expressions associated with the antibiotic exposures. (A) Minimum inhibitory concentrations (MIC) of aztreonam (ATM) alone, clavulanate (CA) alone, and aztreonam–clavulanate (ATM/CA) combination against the three strains of *E. coli*. (B) The PCA plot based on gene expression of 0.5 mg/L aztreonam (ATM) exposure alone, 0.5 mg/L clavulanate (CA) exposure alone, 0.5/0.5 mg/L aztreonam–clavulanate (ATM/CA) combination exposure, and control. (C) The heat map of the top 1,000 differentially expressed genes in the 0.5 mg/L aztreonam (ATM) exposure, 0.5 mg/L clavulanate (CA) exposure, 0.5/0.5 mg/L aztreonam/clavulanate (ATM/CA) combination exposure, and the control. (D) Differential gene expression upon 0.5 mg/L aztreonam exposure alone vs. control. The dashed lines represent the threshold of extremely significantly up- or down-regulated genes ($|\log_2\text{FoldChange}| > 1$ and padj ≤ 0.05; padj is the adjusted *p* value using Benjamin–Hochberg correction). Purple, red, and gray dots represent significantly up-regulated, down-regulated, and not significantly differentially expressed genes, respectively. (E) Differential gene expression upon 0.5 mg/L clavulanate exposure alone vs. control. Other details are the same as in (C). (F) Differential gene expression upon 0.5/0.5 mg/L aztreonam–clavulanate combination exposure vs. control. Other details are the same as in (C).

**TABLE 2** MIC of aztreonam (ATM) alone, clavulanate (CA) alone, and aztreonam/clavulanate (ATM/CA) combination against *E. coli* clinical isolates, as well as β-lactamase genes encoded in each isolate[a]

| Strains | β-Lactamase genes encoded | MIC (mg/L) | | |
|---|---|---|---|---|
| | | ATM | CA | Atm/CA |
| *E. coli* ATCC 25922 | | 0.25 | -[b] | - |
| *E. coli* ATCC 35218 | | 0.25 | - | - |
| *E. coli* ymmD45 | $bla_{NDM-13}$, $bla_{TEM-1B}$, $bla_{TEM-141}$, $bla_{CTX-M-55}$, and $bla_{OXA-1}$ | 64 | 32 | 0.5/0.5 |

[a]ATM, aztreonam; CA, clavulanate.
[b]"-" Indicates that no MIC is measured.

elicited increased transmembrane transporter activity, chemotaxis responses, and flagellum-dependent cell motility (Fig. S4A). KEGG analysis revealed active expression of ATP-binding cassette (ABC) transporter systems, two-component signaling systems (TCS), and flagellar assembly processes (Fig. S4B). Similar to aztreonam exposure, low-concentration clavulanate exposure resulted in a significant reduction in overall cytoplasmic activity, ribosome assembly, and RNA binding processes (Fig. S5).

Taken together, these findings collectively suggest that both aztreonam and clavulanate, when applied individually, exert inhibitory effects on the metabolism, membrane integrity, and CAMP resistance of *E. coli* ymmD45 to varying degrees. However, *E. coli* ymmD45 demonstrates a capacity to overcome these inhibitions through increased transporter activity, enhanced membrane assembly, and accelerated flagellar motility. It is noteworthy that both aztreonam and clavulanate, when administered individually, led to the upregulation of genes associated with membrane functions in *E. coli* ymmD45. In the case of aztreonam exposure, this observation can be attributed to aztreonam's ability to impede the final stages of bacterial membrane synthesis by inhibiting penicillin-binding proteins, which elucidates the upregulation of membrane-related genes. In contrast, clavulanate, while causing limited gene expression changes, demonstrates restricted antibacterial activity when not combined with a partnering β-lactam antibiotic.

## Aztreonam–clavulanate combination significantly inhibited *E. coli* ymmD45

The concurrent administration of clavulanate and aztreonam reveals substantial synergistic effects when combating multidrug-resistant *E. coli* ymmD45, as shown in Table 1. To unravel the underlying impact of the aztreonam–clavulanate combination on *E. coli* ymmD45, we also conducted RNAseq analyses following exposure with aztreonam and clavulanate at a concentration of 0.5 mg/L each. PCA and heatmaps were employed to ensure sample-level quality control (Fig. 3B and C; Table S4). After the aztreonam–clavulanate combination exposure, we identified 1,023 up-regulated and 685 down-regulated genes in comparison to the control (Fig. 3F; Table S8). This combined exposure revealed a broader spectrum of differentially expressed genes and associated downstream information.

When compared to the control, GO analysis revealed the significant upregulation of genes linked to the integral components of the cellular membrane and plasma membrane, transmembrane transporter activity, kinase activity, oxidoreductase activity, and oxidoreductase activity-related processes. In parallel, KEGG analysis indicated a remarkable elevation in ABC transporters and flagellar assembly processes (Fig. 4A). Conversely, the aztreonam–clavulanate combined exposure resulted in a significant reduction in expression of genes associated with the cytosol, cytoplasm, protein/RNA binding, regulation of transcription (DNA-templated), and cytoplasmic translation (Fig. 4B).

We have successfully discerned 19 up-regulated and 120 down-regulated DEGs when exposed to aztreonam–clavulanate combination vs. aztreonam exposure alone (Fig. S6A; Table S9), as well as 16 up-regulated and 229 down-regulated DEGs in comparison to clavulanate exposure alone (Fig. S6B; Table S10). The genes with significant changes

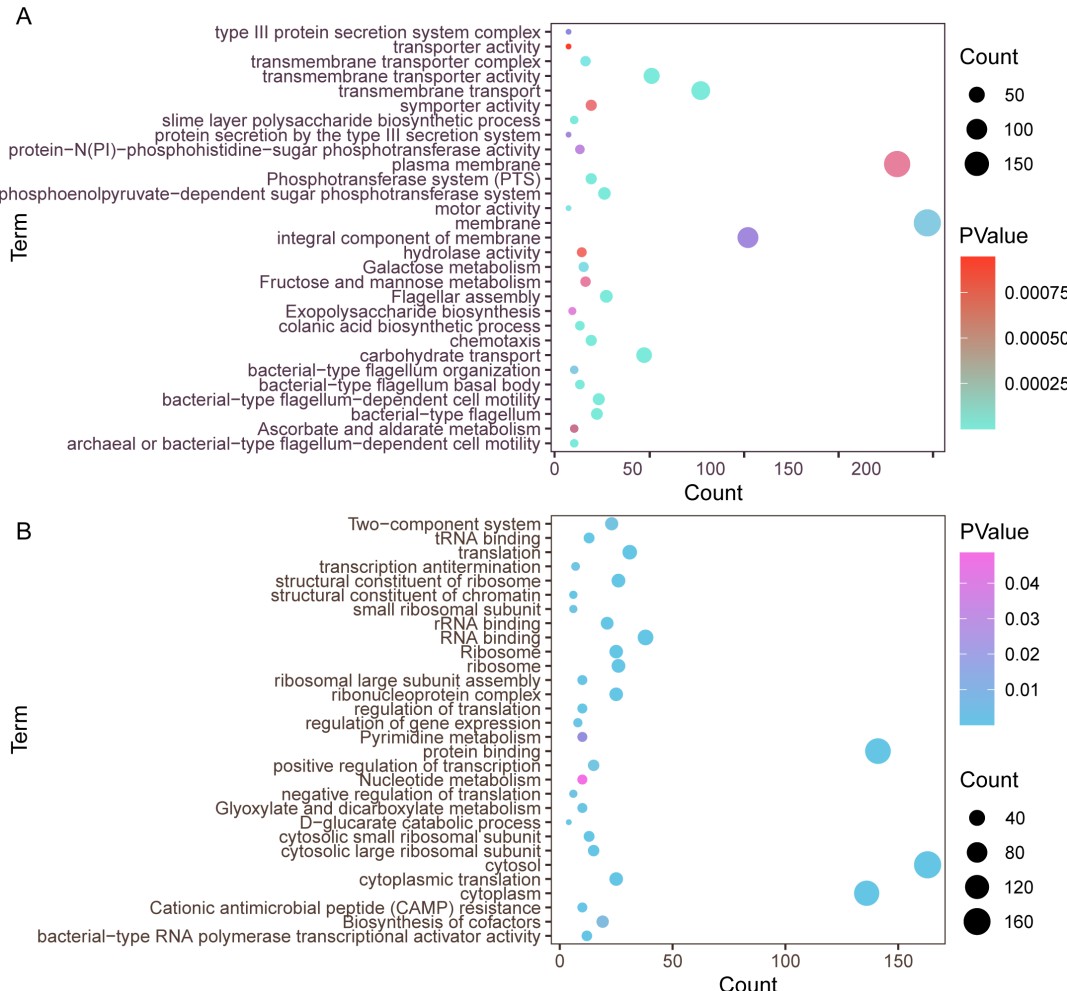

**FIG 4** GO and KEGG analyses on DEGs from 0.5/0.5 mg/L aztreonam–clavulanate combination exposure vs. control. (A) GO and KEGG analyses on up-regulated DEGs from 0.5/0.5 mg/L aztreonam–clavulanate combination exposure vs. control. (B) GO and KEGG analyses on down-regulated DEGs from 0.5/0.5 mg/L aztreonam–clavulanate combination exposure vs. control.

allow for a more in-depth exploration of the hidden mechanisms behind the aztreonam–clavulanate combination. When compared to the aztreonam exposure alone, aztreonam–clavulanate combination exposure up-regulated ATP binding activities, tricarboxylic acid cycle, sulfur metabolism, biosynthesis of secondary metabolites, and cellular response to stimulus and down-regulated many catabolic processes, cellular response, and outer membrane under the aztreonam–clavulanate combination exposure (Fig. 5). According to protein–protein interaction (PPI) network analysis on the differentially expressed genes, functions, such as protein transporting, protein targeting and membrane organization, were inhibited, and sulfur metabolism and glyoxylate bypass were up-regulated in the gene network (Fig. 6A and B). While compared to the clavulanate exposure alone, aztreonam–clavulanate combination exposure up-regulated organic substance metabolic process and catalytic activity and down-regulated many activities, mainly including membranes in different locations, bacterial-type flagellum assembly and activity, and ABC transporters based on the PPI network analysis (Fig. 6C and D; Fig. 7).

Further examination focused on the top 10 up-regulated and 10 down-regulated genes compared to aztreonam or clavulanate exposure alone (Table S11; sorted by FoldChange from RNAseq analysis, $p < 0.05$). Among the top 10 up-regulated DEGs, *phoQ* is a constituent of the PhoP–PhoQ TCS, and *gpsA* is involved in membrane lipid metabolism. Conversely, among the top 10 down-regulated DEGs, *yjbG* belongs to a class of

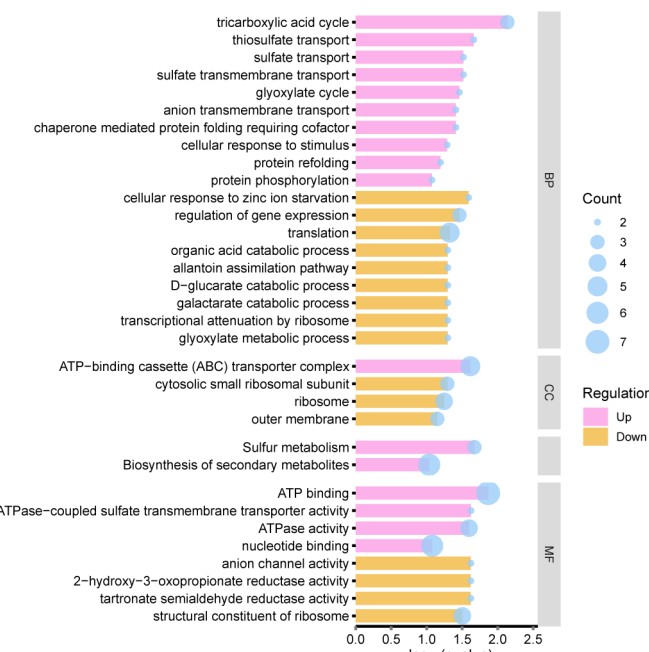

**FIG 5** GO and KEGG analyses on up-regulated and down-regulated DEGs in 0.5/0.5 mg/L aztreonam–clavulanate combination exposure vs. 0.5 mg/L aztreonam exposure alone. BP, CC, and MF represent biological process, cellular component, and molecular function, respectively.

stress genes strongly induced by ethanol treatment (52); *yobH* is involved in the invasion of host cells in *Salmonella typhimurium* (53); and *iss* has long been recognized for its role in extraintestinal pathogenic *E. coli* virulence and has been localized to large virulence plasmids (54). Indeed, in contrast to the single action of the two antibiotics, the synergistic application of aztreonam and clavulanate demonstrated a notable reduction in bacterial resilience against external stress, a diminished capacity to infect the host, and an effective mitigation of bacterial virulence. To conclude, the effectiveness of the aztreonam–clavulanate combination in achieving pathogen inhibition can be attributed to its potent synergy, which not only inhibits the expression of key virulence genes but also disrupts two-component regulatory systems. Moreover, it interferes with the complete formation of the cellular membrane by two phases, culminating in a multifaceted approach to bacterial inhibition.

## Rapid acquisition of resistance mutations upon single-step selection with aztreonam–clavulanate combination

The development of resistance to aztreonam–clavulanate could arise due to the selection of cells carrying resistance mutations and/or other advantageous traits conducive to clonal expansion. In our investigation, we screened a total of 80 resistant colonies on LB agar plates containing 8 mg/L for aztreonam and clavulanate (Fig. 1). Of these, 71 samples met the criteria for inclusion in the final mutation analysis. Collectively, our analysis revealed the presence of 24 BPSs and one 8 bp insertion in 15 strains (Table S12). We then proceeded to conduct MIC tests in LB broth on these 15 mutants (Table S12). To note, the resistance of all 15 mutants to aztreonam/clavulanate had increased to varying degrees. The lower MIC values compared to the 8 mg/L on LB agar plates for single-step mutant screening are due to the fact that MIC tests were conducted in liquid medium, where bacterial cells are fully exposed to the antibiotics. In contrast, our screening on LB agar plates was performed on solid agar, which limits the contact between cells and the agar surface, resulting in incomplete exposure and consequently higher MIC values.

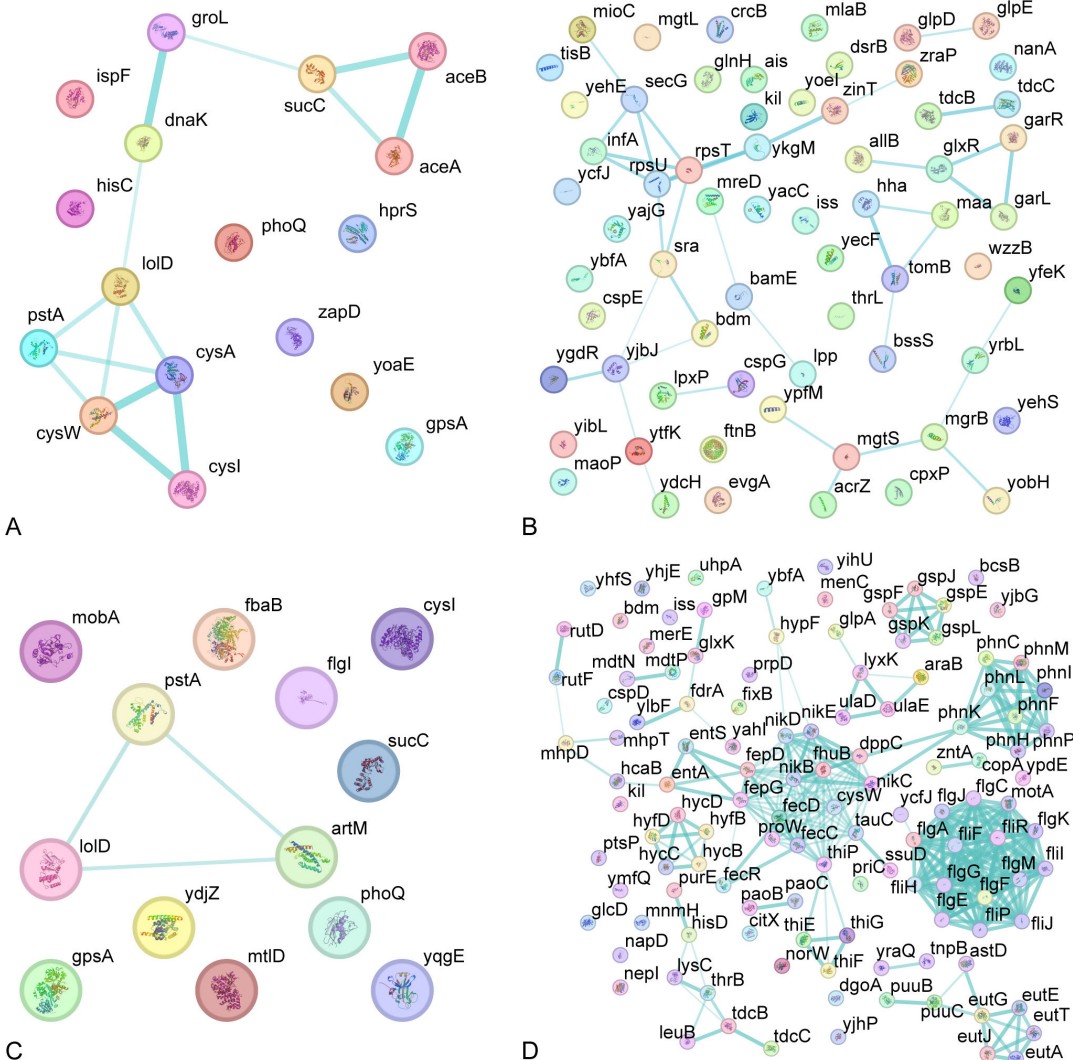

**FIG 6** Protein–protein interaction network of DEGs. Each node represents all proteins produced by a single protein-coding gene locus. Colored nodes represent query proteins and first shell of interactors. The thickness of lines indicates edge confidence. (A) Protein–protein interaction network of up-regulated DEGs in 0.5/0.5 mg/L aztreonam–clavulanate combination exposure vs. 0.5 mg/L aztreonam exposure alone. (B) Protein–protein interaction network of down-regulated DEGs in 0.5/0.5 mg/L aztreonam–clavulanate combination exposure vs. 0.5 mg/L aztreonam exposure alone. (C) Protein–protein interaction network of up-regulated DEGs in 0.5/0.5 mg/L aztreonam–clavulanate combination exposure vs. 0.5 mg/L clavulanate exposure alone. (D) Protein–protein interaction network of down-regulated DEGs in 0.5/0.5 mg/L aztreonam–clavulanate combination exposure vs. 0.5 mg/L clavulanate exposure alone.

Among these mutations, the G:C→A:T transitions were the most abundant mutation type, and the transition/transversion ratio is 1.67. Such mutation spectrum is highly consistent with that reported in spontaneous mutation accumulation studies of *E. coli* (55–57). The results thus inferred the spontaneous origin of these resistance mutations. Most mutations fell in nine genes that could be annotated (Fig. 8). The mutated genes were subjected to GO analysis, revealing associations with the regulation of initiation of DNA-templated transcription, cytosolic DNA-directed RNA polymerase complex, transcription-factor activity, and sequence-specific DNA binding. These pathways exhibit significant overlap with those identified in the RNAseq analysis comparing aztreonam–clavulanate combination exposure to other exposures (Table S13). Furthermore, the *narK* gene is consistently implicated in nearly every GO term related to membrane composition and transmembrane-transport activities (58). This underscores its significance in influencing these cellular processes.

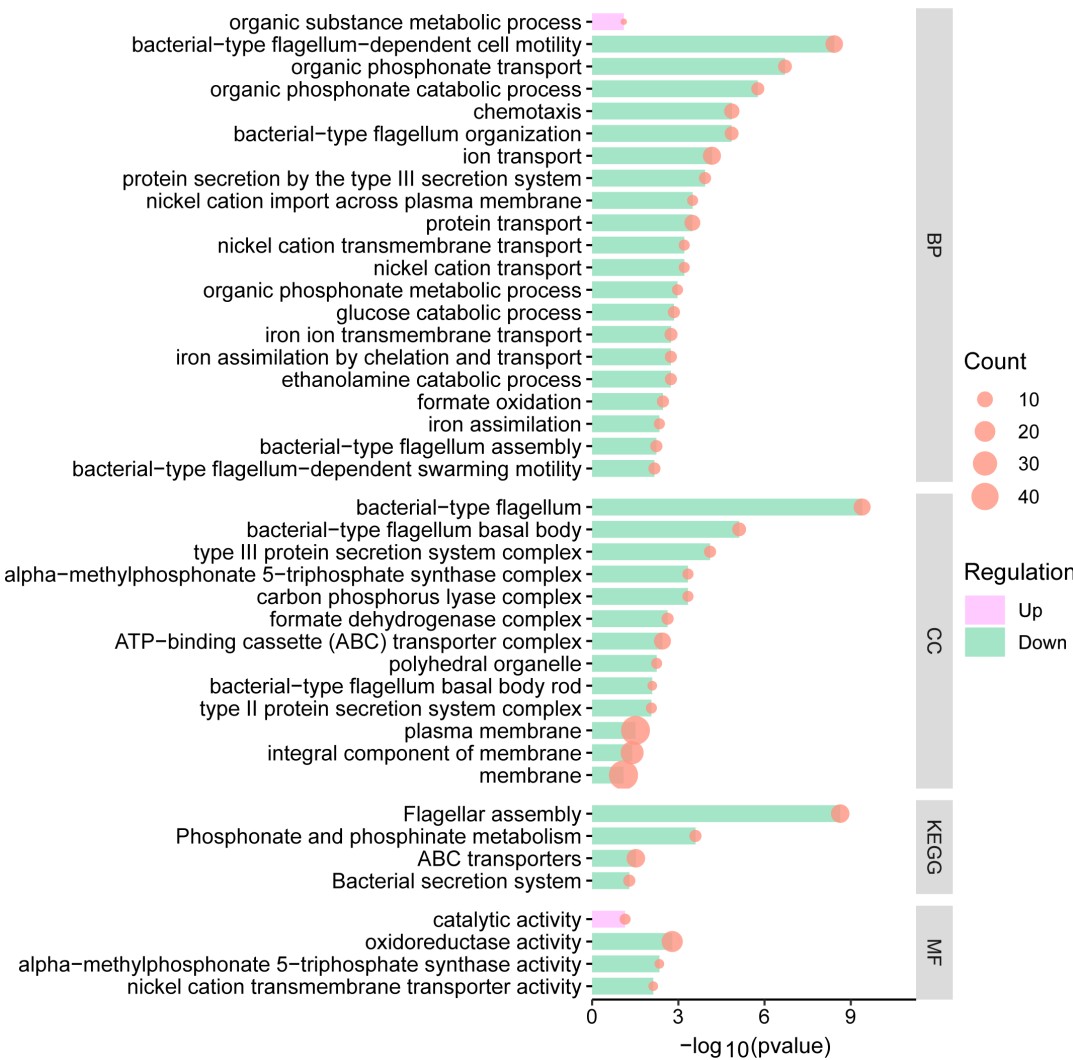

**FIG 7** Pathway enrichment analyses on up-regulated and down-regulated DEGs from 0.5/0.5 mg/L aztreonam–clavulanate combination exposure vs. 0.5 mg/L clavulanate exposure alone. BP, CC, and MF represent biological process, cellular component, and molecular function, respectively.

These genes enriched with mutations encompass *rpoD* and *rpoS*, which are pivotal sigma-factor genes (59), with the *rpoS* gene playing a crucial role in bacterial stress response and adaptation to adverse environmental conditions (60); *fis* is responsible for encoding the global regulatory protein Fis (factor for inversion stimulation) in *E. coli* (61) (Table S12). These observations reveal that, in the context of acute high concentrations of aztreonam–clavulanate exposure, the emergence of populational resistance to the aztreonam–clavulanate combination predominantly associates with osmoregulation and the regulation of transcription metabolism, cellular growth, and motility, which is consistent with the results of the transcriptome analysis above.

## DISCUSSION

Infections attributed to MBL-producing gram-negative bacteria present a formidable therapeutic challenge characterized by limited treatment options and a significant associated mortality rate exceeding 30% (62, 63). Notably, *E. coli* stands as one of the most prevalent gram-negative pathogens in clinical settings, and its escalating antibiotic resistance poses a substantial threat to public health (64). Our previous studies have demonstrated the synergistic effects of aztreonam in combination with clavulanate

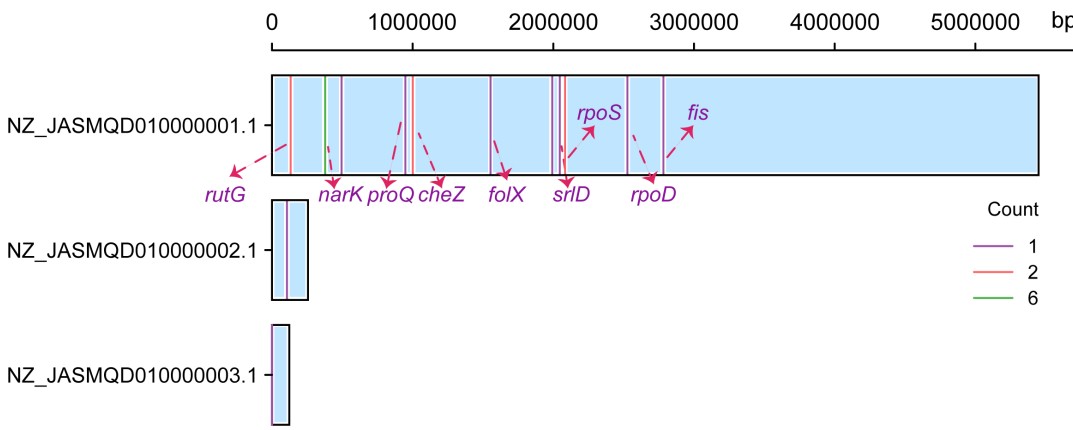

**FIG 8** Distribution of base pair substitution mutations across the *E. coli* ymmD45 contigs.

on *E. coli* carrying the NDM resistance gene (65). The present study adopts an integrated approach, including differential gene expression, gene enrichment, gene network analyses, and single-step screening of resistance mutants combined with deep whole-genome sequencing, to probe the mechanisms underlying the actions of aztreonam and clavulanate against MDR *E. coli*. Our findings imply the spontaneous origin of the resistance mutations during the aztreonam–clavulanate combination treatment and substantiate the strong inhibitory potential of the aztreonam–clavulanate combination against MDR *E. coli*. Specifically, aztreonam emerges as a key player in inhibiting cell wall synthesis, thereby impeding bacterial growth and reproduction. Clavulanate amplifies and prolongs this inhibitory effect while also inducing disruptions in the two-component regulatory system and RNA binding activity, among other critical cellular processes.

MDR strains containing NDM genes potentially exhibit polygenic resistance. The effective treatment of *E. coli* ymmD45 using aztreonam–clavulanate combination suggests that the notable pathogen inhibition arises from a polygenic basis, wherein each gene exerts a modest yet cumulative impact. The aztreonam–clavulanate combination effectively inhibits the growth of *E. coli* ymmD45 by exerting influence on several pivotal genes, namely, *rstA*, *iss*, *cyoA*, and *fimA*, all of which play crucial roles in the intracellular regulatory system, host invasion, metabolic processes, membrane integrity, and motility. Notably, RstA is a recognized regulator within the TCS (RstAB) and functions as a global virulence regulator in *E. coli* (66). The notable down-regulation of the *rstA* gene under the influence of the aztreonam/clavulanate combination significantly impacts the bacterial adsorption capacity, tolerance, toxin production, and motility. CyoA (subunit II) represents a crucial inner membrane protein characterized by its dual-spanning membrane topology and the presence of a cleavable signal peptide (67, 68). Simultaneously, the *fimA* gene encodes the type-1 fimbrial subunit of *E. coli* (69). The substantial down-regulation of these genes precipitates disturbances in the overarching regulatory and metabolic systems of the bacteria, leading to structural perturbations in the cellular membrane and a notable attenuation of motility.

In pathogenic bacteria, the two-component regulatory systems typically serve as basic stimulus–response coupling mechanisms to allow organisms to sense a specific environmental stimulus and respond to changes in many different environmental conditions, playing important roles in signal transduction and regulation of pathogenesis (70, 71). The PhoP–PhoQ TCS plays a role in the virulence properties of a number of bacterial species, and the RstA–RstB TCS impacts virulence, motility, cell morphology, and penicillin tolerance of *Photobacterium damselae* subsp. *damselae* (72–74). Previous studies have shown that the PhoP–PhoQ and RstA–RstB systems are coordinated with regard to the regulation of specific proteins as well as metabolic processes (75, 76). Aztreonam–clavulanate combination inhibits bacterial growth by attacking these two

master TCSs, resulting in disruption of intracellular metabolism, reduced surface motility, invasion, and virulence.

*fis* and *proQ* are known to function in bacterial life cycles and to be associated with bacterial virulence. *fis* is known as a transcriptional regulator for an amount of genes (77). In *E. coli*, Fis is the most abundant nucleoid-associated protein during the exponential growth phase in rapidly growing cultures (78). Recent evidence suggests that Fis is a versatile regulator, which optimizes bacterial virulence responses to particular challenges in several pathogenic bacteria (79, 80). ProQ impacts virulence and stress–response gene expression in bacterial pathogenesis as a major RNA-binding protein in *E. coli* (81). In this study, *proQ* showed a significantly down-regulated trend in the clavulanate exposure, and *fis* showed significant down-regulation in the aztreonam exposure. Only when exposed to a combination of aztreonam and clavulanate were both genes down-regulated at the same time. Therefore, this indicates that aztreonam–clavulanate combination can effectively inhibit MDR pathogenic bacteria; even if some bacteria can survive, survival is at the cost of a significant reduction in virulence.

It is imperative to acknowledge the limitations in this study. While we successfully identified numerous genes associated with the effective inhibition of MDR bacteria through the application of low-concentration aztreonam–clavulanate combinations, it is important to recognize that certain causal genes were not specified within our analysis, as well as the molecular dynamics of the synergy. Another limitation is that this study is conducted on only one strain. To address these limitations, future endeavors requiring a significant workload are needed, including more species and strains resistance testing, functional assessments, and gene editing for each pertinent gene of interest.

In conclusion, our analysis reveals the intracellular activity of the aztreonam–clavulanate combination against MDR *E. coli*. This study provides insights into the underlying mechanisms contributing to its enhanced bactericidal efficacy. *In vitro* exposure to the aztreonam–clavulanate combination of *E. coli* leads to profound membrane disruption, affecting the PhoP–PhoQ and RstA–RstB TCSs, as well as metabolic processes, thereby potentially contributing to an extended bactericidal effect. This notable inhibitory effect of the aztreonam–clavulanate combination arises from its collective suppression of several key genes, especially *fis*, *proQ*, *iss*, *rstA*, *cyoA*, and *fimA*, resulting in a significant reduction in the toxicity and virulence of pathogenic bacteria while inhibiting their growth. This antibiotic combination holds significant clinical implications, as it contributes to the reduction of virulence and pathogenic bacterial density, even at minimal antibiotic concentrations, offering a novel strategy for clinical drug utilization. While these findings are promising, it is important to note that they are based on *in vitro* studies. Further research, including *in vivo* studies and clinical trials, is necessary to fully understand the clinical implications and efficacy of these combinations. Exploring such combinations could eventually contribute to novel treatment strategies, especially in combating multidrug-resistant bacterial infections. Our study aims to facilitate further research on antibiotic combinations and their effects on bacterial physiology. By doing so, we hope to support the development of more targeted and effective treatment strategies, ultimately improving patient outcomes and informing antibiotic use in clinical practice.

## ACKNOWLEDGMENTS

We appreciate the technical help from the maintenance team of the IEMB-1 computation clusters at OUC.

This study was supported by the National Natural Science Foundation of China (32270435, 32471688), Taishan Scholars Program of Shandong Province, the National Institutes of Health MIRA grant (2R35GM122566), and Wuhu City Application and Fundamental Research Project (2022jc08).

## AUTHOR AFFILIATIONS

[1]Key Laboratory of Evolution & Marine Biodiversity (Ministry of Education) and Institute of Evolution & Marine Biodiversity, School of Medicine and Pharmacy, Ocean University of China, Qingdao, Shandong Province, China

[2]Laboratory for Marine Biology and Biotechnology, Laoshan Laboratory, Qingdao, Shandong Province, China

[3]College of Life Sciences, Anhui Normal University, Wuhu, Anhui Province, China

[4]Department of Statistics, State University of Maringá, Maringá, Paraná, Brazil

[5]Biodesign Center for Mechanisms of Evolution, Arizona State University, Tempe, USA

## AUTHOR ORCIDs

Tongtong Lin  http://orcid.org/0000-0002-3515-2018
Sherwin K. B. Sy  http://orcid.org/0000-0001-6405-9130
Hongan Long  http://orcid.org/0000-0001-9767-8173
Mingming Yu  http://orcid.org/0000-0001-9504-891X

## AUTHOR CONTRIBUTIONS

Tongtong Lin, Data curation, Formal analysis, Methodology, Visualization, Writing – original draft, Writing – review and editing | Jiayuan Zhang, Investigation, Methodology, Writing – review and editing | Shuo Diao, Methodology, Writing – review and editing | Jinke Yan, Methodology, Validation, Writing – review and editing | Kexin Zhang, Methodology, Writing – review and editing | Jichao Cao, Methodology, Writing – review and editing | Junyi Huang, Methodology, Validation, Writing – review and editing | Yaohai Wang, Visualization, Writing – review and editing | Zhihua Lv, Supervision, Writing – review and editing | Xiaopeng Shen, Funding acquisition, Writing – review and editing | Sherwin K. B. Sy, Writing – review and editing | Michael Lynch, Funding acquisition, Writing – review and editing | Hongan Long, Conceptualization, Funding acquisition, Supervision, Writing – review and editing | Mingming Yu, Project administration, Writing – review and editing

## DATA AVAILABILITY

The *de novo* assembled genome and the annotation files of *E. coli* ymmD45 were uploaded to NCBI GenBank: JASMQD010000000. All raw whole-genome sequences and RNAseq raw data were deposited in NCBI SRA with BioProject PRJNA976513.

## ADDITIONAL FILES

The following material is available online.

### Supplemental Material

**Supplemental Figures (Spectrum01782-24-s0001.pdf).** Figures S1 to S6.
**Supplemental Tables (Spectrum01782-24-s0002.xlsx).** Tables S1 to S13.

### Open Peer Review

**PEER REVIEW HISTORY (review-history.pdf).** An accounting of the reviewer comments and feedback.

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
