## [Reviewer comments · Microbiology Spectrum]

Microbiology Spectrum

The impact of aztreonam-clavulanic acid exposure on gene expression and mutant selection using a multidrug resistant *E. coli*

Tongtong Lin, Jiayuan Zhang, Shuo Diao, Jinke Yan, Kexin Zhang, Jichao Cao, Junyi Huang, Yaohai Wang, zhihua lv, Xiaopeng Shen, Sherwin Sy, Michael Lynch, Hongan Long, and Mingming Yu

Corresponding Author(s): Mingming Yu, Ocean University of China - Yushan Campus

Review Timeline:

Submission Date:	August 4, 2024
Editorial Decision:	November 4, 2024
Revision Received:	November 10, 2024
Editorial Decision:	November 12, 2024
Revision Received:	December 12, 2024
Accepted:	January 6, 2025

Editor: Krisztina Papp-Wallace

Reviewer(s): The reviewers have opted to remain anonymous.

Transaction Report:

DOI: <https://doi.org/10.1128/spectrum.01782-24>

Re: Spectrum01782-24 (The molecular mechanisms and resistance mutagenesis in the synergistic action of aztreonam and clavulanate against multidrug-resistant *Escherichia coli*)

Dear Prof. Mingming Yu:

Thank you for the privilege of reviewing your work. Below you will find my comments, instructions from the Spectrum editorial office, and the reviewer comments.

I am pleased to inform you that your manuscript has been editorially accepted for publication. However, there are a few additional questions in the submission form that need to be answered before the final decision. Once these are completed, please return your submission so that I can move the paper forward to acceptance.

Revision Guidelines

Sincerely,
Krisztina Papp-Wallace
Editor
Microbiology Spectrum

Re: Spectrum01782-24R1 (The molecular mechanisms and resistance mutagenesis in the synergistic action of aztreonam and clavulanate against multidrug-resistant *Escherichia coli*)

Dear Prof. Mingming Yu:

Thank you for the privilege of reviewing your work. Below you will find my comments, instructions from the Spectrum editorial office, and the reviewer comments.

After reviewing the revised manuscript, there are several areas that need to be addressed in the manuscript. See comments below and attached marked up manuscript for additional feedback.

Comments:

1. Revise the title: The impact of aztreonam-clavulanic acid exposure on gene expression and mutant selection using a multidrug resistant *E. coli*.
2. Caution: Clavulanic acid doesn't have antimicrobial activity on its own, thus stating that in combination with aztreonam is "synergistic" is not correct. The combination is effective against the MDR strain of *E. coli* in this manuscript. Aztreonam is not hydrolyzed by metallo-beta-lactamases, however, can be hydrolyzed by serine beta-lactamases with an expanded profile, thus clavulanic acid is needed to inhibit those serine beta-lactamases. That is why the combination is effective against the ymmD45 strain of *E. coli*. This isn't clear in the manuscript as currently written. Lines 28-30 are not accurate. We do understand why this combination is effective against this strain. The manuscript is largely focused on the impact of antibiotic exposure on gene expression, which is a distinct experiment from assessing the mechanism of action of the antibiotic. This needs to be clarified in the paper.
3. The focus of the paper should be on transcriptomic analysis conducted on the strain exposed to aztreonam and clavulanic acid and the single-step mutant selection.
4. Line 33, you aren't really treating the cells with aztreonam and clavulanic acid for the transcriptomic analysis, you are exposing them to the compounds.
5. Did exposing the cells to MIC concentrations of compound impact on growth? Were growth curves obtained?
6. Like 103, the mechanism of action is the aztreonam inhibits PBP3 and is not hydrolyzed by NDM, while clavulanic acid inhibits serine beta-lactamases, allowing aztreonam to inhibit PBP3.
7. Line 131, how much clavulanic acid was used for MIC testing? Was it held a fixed concentration as BLIs are typically evaluated?
8. Line 136, FICI should only be obtained for two antibiotics. Clavulanic acid is not an antibiotic. Please remove FICI from the manuscript.
9. Line 139, the approach used is a single-step mutant screen, were the mutants evaluated for stability (i.e., streaked onto LB agar only 3x and back onto the LB agar w/drugs and it still grew)? What were the MICs for the mutants, this should be included in the paper.
10. Line 264, list all beta-lactamase genes that are present in the strain and use the conventional naming scheme. For NDM-1 gene, it's blaNDM-1
11. Line 265, clavulanic acid does not have antimicrobial activity against *E. coli*. There are no CLSI breakpoints for clavulanic acid alone. Please revise.

Revision Guidelines

For complete guidelines on revision requirements, see our Submission and Review Process webpage. Submission of a paper

that does not conform to guidelines may delay acceptance of your manuscript.

Sincerely,
Krisztina Papp-Wallace
Editor
Microbiology Spectrum

12 Dec. 2024

RE: Spectrum01782-24R1

Dear Editor,

We truly appreciate the opportunity to revise our paper previously titled “The molecular mechanisms and resistance mutagenesis in the synergistic action of aztreonam and clavulanate against multidrug-resistant *Escherichia coli*”. Based on the feedback, we have revised the manuscript and addressed all the comments carefully. Please find the revised manuscript with changes highlighted, and our responses to all comments are as shown below.

Best regards,

Mingming Yu, on behalf of all authors

1. Revise the title: The impact of aztreonam-clavulanic acid exposure on gene expression and mutant selection using a multidrug resistant *E. coli*.

RE: Done as suggested.

2. Caution: Clavulanic acid doesn't have antimicrobial activity on its own, thus stating that in combination with aztreonam is "synergistic" is not correct. The combination is effective against the MDR strain of *E. coli* in this manuscript. Aztreonam is not hydrolyzed by metallo-beta-lactamases, however, can be hydrolyzed by serine beta-lactamases with an expanded profile, thus clavulanic acid is needed to inhibit those serine beta-lactamases. That is why the combination is effective against the ymmD45 strain of *E. coli*. This isn't clear in the manuscript as currently written. Lines 28-30 are not accurate. We do understand why this combination is effective against this strain. The manuscript is largely focused on the impact of antibiotic exposure on gene expression, which is a distinct experiment from assessing the mechanism of action of the antibiotic. This needs to be clarified in the paper.

3. The focus of the paper should be on transcriptomic analysis conducted on the strain exposed to aztreonam and clavulanic acid and the single-step mutant selection.

RE: Thanks for the insights. We have removed lines 28-30 and revised other relevant sentences in the manuscript to clarify that the paper is focusing on elucidating the impact of aztreonam and/or clavulanic acid exposure on gene expression through transcriptome analysis conducted on the clinically isolated strain *E. coli* ymmD45 and the single-step mutant selection.

4. Line 33, you aren't really treating the cells with aztreonam and clavulanic acid for the transcriptomic analysis, you are exposing them to the compounds.

RE: We have now revised this sentence to “In this investigation, we conducted transcriptomic analysis to reveal the downstream differential gene expression in *E. coli* ymmD45 (a strain newly isolated and found to carry the New Delhi metallo- β -lactamase gene) following exposure to

aztreonam and clavulanate separately, as well as their combination.” We have also made changes to other parts of the manuscript with inaccurate statements.

5. Did exposing the cells to MIC concentrations of compound impact on growth? Were growth curves obtained?

RE: A sterile 96-well microdilution plate was used to determine the MIC of aztreonam, clavulanate and aztreonam/clavulanate combination against the clinical isolate *E. coli* ymmD45. *E. coli* ymmD45 could grow when exposed to MIC concentrations. Time-kill assay was performed when exposed to 1× MIC, and time-kill curves were obtained, as shown below. The exposure to MIC concentrations of aztreonam, clavulanate, and aztreonam/clavulanate combination did inhibit growth compared with the negative control.

6. Line 103, the mechanism of action is the aztreonam inhibits PBP3 and is not hydrolyzed by NDM, while clavulanic acid inhibits serine beta-lactamases, allowing aztreonam to inhibit PBP3.

RE: We have now revised this sentence to “To elucidate the impact of the aztreonam-clavulanate combination, we performed RNAseq-based differential gene-expression, pathway-enrichment and gene network analyses of *E. coli* ymmD45 following exposure to them.”

7. Line 131, how much clavulanic acid was used for MIC testing? Was it held a fixed concentration as BLIs are typically evaluated?

RE: Clavulanic acid MIC was determined at gradient concentrations and did not have a fixed concentration as prescribed by CLSI. The concentration ranges for clavulanic acid was 0.5 to 64 mg/L with a 2-fold step increment for MIC testing. Although clavulanic acid does not have

antimicrobial activity against *E. coli* in theory, it has been found that higher concentrations of clavulanic acid do have antibacterial activity against *E. coli* ymmD45 *in vitro* (Table 2, MIC can be determined).

8. Line 136, FICI should only be obtained for two antibiotics. Clavulanic acid is not an antibiotic. Please remove FICI from the manuscript.

RE: Thanks for the catch. We have removed FICI from the manuscript.

9. Line 139, the approach used is a single-step mutant screen, were the mutants evaluated for stability (i.e., streaked onto LB agar only 3x and back onto the LB agar w/drugs and it still grew)? What were the MICs for the mutants, this should be included in the paper.

RE: We recovered all 80 resistant colonies from previously frozen samples, streaked them three times onto LB agar only and then streaked onto LB agar plates with 8 mg/L of aztreonam/clavulanate (16× MIC). Upon incubation, we observed that only 15 strains, exactly the ones with BPSs or indel mutations, were capable of growth under these conditions, demonstrating that our single-step screening experiment combined with the Illumina whole-genome deep sequencing method is highly effective.

We then proceeded to conduct MIC tests in LB broth on these 15 mutants (results are shown below and also added to Supplementary Table S12). To note, the resistance of all 15 mutants to aztreonam/clavulanate had increased to varying degrees. The lower MIC values compared to the 8 mg/L on LB agar plates for single-step mutant screening are because MIC tests were conducted in liquid medium, where bacterial cells were fully exposed to the antibiotics. In contrast, our screening on LB agar plates was performed on solid agar, which limits the contact between cells and the agar surface, resulting in incomplete exposure and consequently higher MIC values. We have added such results in relevant descriptions in the manuscript.

Strains	MIC (mg/L)			Strains	MIC (mg/L)		
	ATM	CA	ATM/CA		ATM	CA	ATM/CA
ymmD45	64	32	0.5/0.5	B6	>64	64	4/4
A1	64	32	1/0.5	B8	64	32	0.5/2
A2	64	32	2/0.5	B9	>64	32	1/2
A3	>64	32	1/2	C2	64	32	1/1
A4	64	32	4/2	C4	64	32	2/2
A5	64	32	1/2	C11	>64	32	1/2
A6	>64	32	1/2	G22	32	32	2/1
B1	>64	32	1/1	G24	64	32	2/1

10. Line 264, list all beta-lactamase genes that are present in the strain and use the conventional naming scheme. For NDM-1 gene, it's blaNDM-1.

RE: We have now revised the sentence to “Notably, *E. coli* ymmD45 is host to an array of resistance genes, including the *bla*_{NDM-13}, *bla*_{TEM-1B}, *bla*_{TEM-141}, *bla*_{CTX-M-55} and *bla*_{OXA-1}.” We have also revised other parts of the manuscript correspondingly.

11. Line 265, clavulanic acid does not have antimicrobial activity against *E. coli*. There are no CLSI breakpoints for clavulanic acid alone. Please revise.

RE: We have now revised the sentence to “The MICs for control strains *E. coli* ATCC 25922 and ATCC 35218 were both found to be below 0.5 mg/L for aztreonam (Fig. 1 and Fig. 3A; Table 2). These tests met the CLSI (the Clinical and Laboratory Standards Institute) standards for accuracy, reliability, and reproducibility in laboratory testing.”

Re: Spectrum01782-24R2 (The impact of aztreonam-clavulanic acid exposure on gene expression and mutant selection using a multidrug resistant *E. coli*)

Dear Prof. Mingming Yu:

Your manuscript has been accepted, and I am forwarding it to the ASM production staff for publication. Your paper will first be checked to make sure all elements meet the technical requirements. ASM staff will contact you if anything needs to be revised before copyediting and production can begin. Otherwise, you will be notified when your proofs are ready to be viewed.

Sincerely,
Krisztina Papp-Wallace
Editor
Microbiology Spectrum